# Ultrasound Assessment of Extracranial Carotids and Vertebral Arteries in Acute Cerebral Ischemia

**DOI:** 10.3390/medicina56120711

**Published:** 2020-12-18

**Authors:** Klearchos Psychogios, Georgios Magoufis, Odysseas Kargiotis, Apostolos Safouris, Eleni Bakola, Maria Chondrogianni, Panagiotis Zis, Elefterios Stamboulis, Georgios Tsivgoulis

**Affiliations:** 1Acute Stroke Unit, Metropolitan Hospital, 18547 Piraeus, Greece; apsychoyio@yahoo.gr (K.P.); kargiody@gmail.com (O.K.); safouris@yahoo.com (A.S.); lstam@med.uoa.gr (E.S.); 2Second Department of Neurology, Attikon University Hospital, 15772 Athens, Greece; elbakola@yahoo.gr (E.B.); mariachondrogianni@hotmail.gr (M.C.); 3School of Medicine, University of Athens, 15772 Athens, Greece; 4Department of Interventional Neuroradiology, Metropolitan Hospital, 18547 Piraeus, Greece; magoufisgeorge1@icloud.com; 5Medical School, University of Cyprus, 1678 Nicosia, Cyprus; takiszis@gmail.com

**Keywords:** carotid doppler ultrasound, stroke, cerebral ischemia

## Abstract

Assessing ischemic etiology and mechanism during the acute phase of an ischemic stroke is crucial in order to tailor and monitor appropriate treatment and determine prognosis. Cervical Duplex Ultrasound (CDU) has evolved since many years as an excellent screening tool for the evaluation of extracranial vasculature. CDU has the advantages of a low cost, easily applicable, bed side examination with high temporal and spatial resolution and without exposing the patients to any significant complications. It represents an easily repeatable test that can be performed in the emergency room as a first-line examination of cervical artery pathology. CDU provides well validated estimates of the type of the atherosclerotic plaque, the degree of stenosis, as well as structural and hemodynamic information directly about extracranial vessels (e.g., subclavian steal syndrome) and indirectly about intracranial circulation. CDU may also aid the diagnosis of non-atherosclerotic lesions of vessel walls including dissections, arteritis, carotid-jugular fistulas and fibromuscular dysplasias. The present narrative review outlines all potential applications of CDU in acute stroke management and also highlights its potential therapeutic implications.

## 1. Introduction

Acute ischemic stroke (AIS) remains one of the leading causes of mortality and morbidity worldwide [1]. Major etiologic subtypes [2] of AIS include cardiac embolism, large artery atherosclerosis, small vessel disease and other less common etiologies (i.e., dissection, angiitis, hematological disorders, etc.). One third of ischemic strokes remain of undetermined cause and a subgroup among them is defined as having embolic strokes of undetermined source (ESUS) following a more granular diagnostic work-up [3]. According to a recent systematic review and metanalysis [4] of the worldwide distribution and temporal trends of AIS etiologic subtypes, large artery atherosclerosis is the most common subtype in Asian patients and the second most common (responsible for almost 20% of AIS) in white populations.

Assessing etiology and mechanism during the acute phase of an ischemic stroke is crucial in order to tailor and monitor appropriate treatment and determine prognosis. Ultrasound has evolved for many years into an excellent screening tool for the evaluation of extracranial and intracranial vasculature that can also sufficiently complement other acute imaging modalities like Magnetic Resonance Imaging/Magnetic Resonance Angiography (MRI/MRA) or computed tomography angiography (CT/CTA). Ultrasound of carotid and vertebral arteries (cervical duplex ultrasonography; CDU) coupled with transcranial doppler sonography are essential parts of the diagnostic workflow in every acute stroke unit, and has some key advantages: high temporal and spatial resolution, real-time evaluation, low cost, bed side application of an exam that can be repeated multiple times throughout hospitalization without exposing the patient to any serious complications [5]. On the other hand, poor reproducibility is the main limitation; it is an operator-dependent examination, thus, adequate education and experience of the performer is a prerequisite. Key questions that can be timely answered by ultrasonography, include:Assessment of ultrasound characteristics and risk of atherosclerotic plaques;Detection and evaluation of approximate degree of underlying stenoses;Diagnosis of occlusion of extracranial and (indirectly) of intracranial arteries;Referral of suitable patients for carotid endarterectomy or carotid artery stenting;Detection of other non-atherosclerotic lesions related to vessel wall pathology in extracranial and intracranial vessels (dissection, fibromuscular dysplasia, Takayasu disease, temporal arteritis, carotid-jugular fistula);Diagnosis of subclavian steal syndrome.

## 2. Carotid Stenosis

CDU includes bilateral examination of common carotid arteries (CCA), internal carotid arteries (ICA), external carotid arteries (ECA) and vertebral arteries (VA). A fast-track neurovascular ultrasound examination [6] with high accuracy in detecting arterial lesions, can be integrated in the hyperacute stroke workflow and facilitate revascularization therapies (Table 1). The typical CDU protocol begins with gray-scale (B-Mode) images of CCA and ICA in transverse and longitudinal planes [7]. These images allow the assessment of the arterial wall, the intima media thickness (IMT) and the identification of atherosclerotic plaques as well as the plaque characteristics: location, length, size, composition, surface of the lesion and estimation of the degree of stenosis or occlusion. Color and power doppler can aid to further delineate the plaque border.

According to a recent update [8] of the Manheim consensus criteria, “IMT is a double-line pattern visualized by echography on both walls of the CCA in a longitudinal image. Two parallel lines, which consist of the leading edges of two anatomical boundaries, form it: the lumen-intima and media-adventitia interfaces”. IMT reflects either an early stage of atherosclerosis or a vascular remodeling in certain conditions that lead to smooth vessel hyperplasia or fibrocellular hypertrophy. It is a priori considered to be an independent predictor of future vascular events [9,10,11] and consequently its value predominantly relates to primary prevention of cardiovascular diseases. Notably, higher CCA-IMT values together with significant carotid stenosis have been linked to an increased risk of recurrence and poor prognosis after a first non-cardioembolic AIS [12,13].

The histopathology of “symptomatic” plaques has demonstrated some well-documented [14,15,16] characteristics linked to plaque rupture and thrombus formation: higher rates of intraplaque hemorrhage, the presence of a lipid-rich necrotic core, thin fibrotic cap with fissuring and ulceration, inflammation and neovascularization. Imaging should aim to identify and display these features. The study of plaque echomorphology may aid to the identification of “unstable” plaques and to the stroke risk stratification especially among patients with mild or moderate stenosis. Plaque morhpology is characterized according to its echolucency (a surrogate of lipid-rich necrotic core) in 5 main categories [17,18]: uniformely echolucent (type I), predominantly echolucent plaques with less than 50% echogenic areas (type II), predominantly echogenic plaques with less than 50% echolucent areas (type III), uniformly echogenic plaques (type IV) and plaques that cannot be classified due to heavy calcification and acoustic shadowing. Symptomatic patients were noted to have predominantly type I and type II lesions. A recent metanalysis [19] showed an almost three times higher risk of recurrence in stroke patients with echolucent compared to echogenic carotid plaques. Hypoechoic plaques are also associated with subcortical and cortical cerebral infarcts of suspected embolic origin, whereas hyperechoic plaques are associated with diffuse white matter infarcts of presumed hemodynamic origin [20]. Computerized semi-automated methods of analyzing plaque features comprise grey-scale median (GSM median of the grey values of all pixels in the plaque image) [21] plaque area and juxta-luminal black (hypoechoeic) areas (JBAs). Several studies [22,23,24] in both symptomatic and asymptomatic patients have correlated lower GSM, increased JBAs and large plaque area with histological features of plaque instability^15^ and future vascular events. An important drawback of ultrasound is the inability to distinguish intraplaque hemorrhage from a lipid-rich necrotic core as both appear hypoechogenic [25].

Carotid plaque surface irregularity assessed by B-mode ultrasound in the Manhattan study, was associated with over a four-fold increased risk of ischemic stroke and a three-fold increased risk among those with carotid stenosis of less than 60% [26]. According to the widely used de Bray criteria [27] a plaque ulcer should measure at least 2 mm in depth and 2 mm in length, with a well-defined back wall at its base when assessed with B-mode sonography, and an area of reversed flow within the recess when assessed with Doppler color-flow imaging (Figure 1). Following the improvement in resolution power and image quality of ultrasound machines, Muraki et al. [28] proposed new criteria stipulating only that “a concavity is clearly visualized in the plaque and that the border echo at the base of the concavity is less intense than the adjacent intimal border of the plaque in the B-mode ultrasound image”. These criteria do not include the findings obtained from Doppler color-flow imaging. However, there is controversy regarding the accuracy of ultrasonography to assess plaque surface and demonstrate irregularities and ulcers with a diagnostic sensitivity ranging from 47% [29] to 95% [30] according to different studies. Contrast-enhanced ultrasound (CEUS) is a promising technique which uses strictly intravascular microbubble contrast agents that can better outline lumen contours and penetrate microvasculature. In a study [31] comparing color Doppler imaging and CEUS with the CT angiography as reference method, CEUS demonstrated higher sensitivity with similar specificity in detecting superficial ulceration. CEUS also shows promising results in highlighting the intraplaque neovascularization [32,33,34], a strong surrogate of the underlying inflammation. However, this method is still not well standardized and its use in clinical practice is limited [35]. A systematic review [36] that summarized the ultrasound characteristics of symptomatic carotid plaques showed that only plaques demonstrating neovascularization, echolucency, intraplaque motion and/or ulceration were associated with symptoms. In the associated metanalysis, plaque neovascularization showed the stronger association with cerebrovascular events (Odds Ratio: 19.68; 95% CI = 3.14–123.16) albeit with a substantial heterogeneity. An uncommon entity that usually presents as an acute emergency is a free-floating thrombus of the common or the internal carotid artery, which is mainly associated with hypercoagulable states [37]. Real-time ultrasound is the best imaging modality in this situation because it can adequately detect the high-risk, pulse-synchronous movement of the thrombus, but also serve as a screening tool for the follow-up of the thrombus evolution.

Even though the plaque characteristics and echomorphology aid in the identification of high-risk patients, in current practice the degree of stenosis remains the main determinant of clinical decisions. According to the Trial of Org 10172 in Acute Ischemic Stroke Treatment (TOAST)^2^ classification, large artery atherosclerosis (LAA) is considered etiologic if there is supportive evidence by CDU or angiography of an occlusion or a stenosis of greater than 50% of an appropriate extracranial or intracranial artery. Moreover, patients with transient ischemic attack (TIA) or mild stroke due to LAA have the highest risk of early recurrence compared to other etiologic subtypes [38] and in case of a carotid stenosis of more than 50% they should be timely (within 3–14 days after the qualifying event) treated with carotid endarterectomy or carotid artery stenting [39,40]. In patients with a TIA or mild stroke and less than 50% stenosis, dual antiplatelet therapy (DAPT) with a loading aspirin and clopidogrel dose (or alternatively ticagrelor) in the first 24 h after presentation, is typically recommended for the first 21–30 days [41]. The US Preventive Services Taskforce (USPSTF) states [42] that CDU remains an excellent non-invasive tool with a sensitivity of 94% and a specificity of 92% for diagnosing 60–99% stenoses.

Standard CDU is also necessary to assess and compare the arteries on each side in order to account for any collateral flow effects. Gray scale (B-Mode) and color flow imaging help to grossly identify a significant stenosis and Spectral Doppler determines velocity. At least two to three spectral analyses of each vessel should be obtained. Analysis should include the stenosis per se as well as the pre- and the post-stenotic area to appreciate indirect hemodynamic effects of the stenosis. Caution should be taken in order to optimize the sampling of the velocity jet, by using the correct angle of insonation (less than 60°). Acoustic shadowing of heavily calcified plaques might also interfere. If the shadowing segment is longer than 2 cm, the degree of stenosis is indeterminate and other imaging modalities are recommended [43]. The reversed flow in the ophthalmic artery (from extracranial to intracranial direction) and similar signs of collateralization of flow (i.e., reversal of A1 segment of Anterior Cerebral Artery, reversal of flow in Posterior Communicating Artery) from the transcranial doppler should also be investigated.

CDU criteria for quantifying the degree of carotid stenosis have been developed through comparisons of velocities derived from Doppler Spectral Waveforms and digital subtraction angiographies [44,45]. According to the Society of Radiologists in Ultrasound Consensus Conference [46] the main parameters assessed for stratification of carotid stenoses, are the ICA Peak Systolic Velocity (ICA PSV), the ICA end diastolic velocity (ICA EDV) and the ICA-to-common-carotid-artery PSV ratio. United Kingdom Joint recommendations [47] further propose peak systolic ICA to end diastolic CCA ratio (referred as St Mary’s ratio) as the more robust index enabling grading in deciles [48]. Table 2 summarizes these criteria. Of note, given that the PSV measured in the CCA may vary along the length of the artery [49], distal CCA measurement should be made within 2 cm of the bifurcation at a point where the vessel still has a uniform diameter. Occlusion of the ICA is present when there is no detectable patent lumen on gray-scale imaging and no flow with spectral, color, and power Doppler (Figure 2). Near occlusion is mainly established by demonstrating a highly stenosed lumen at color or power doppler, because velocity parameters may not apply.

There are also important pitfalls that should be taken into account when interpreting an ultrasound examination. For example, high velocities can be observed due to global velocity increase in hemodilution or anemia, or due to segmental hyperperfusion in the setting of a subarachnoid hemorrhage. Tandem occlusion, contralateral carotid occlusion or occlusion of vertebral arteries, elongated stenoses or high grade stenoses that fall at the other side of the Spencer’s curve, proximal CCA stenosis/occlusions or aortic valve lesions can also be sources of diagnostic uncertainty.

ESUS^3^ has been a timely topic in recent acute stroke literature [50]. The initial hypothesis that the vast majority of ESUS strokes were harboring a cardioembolic source (particularly atrial fibrillation) has been disputed, after the negative results of two randomized controlled clinical trials that compared the use of Novel Oral Anticoagulants to Aspirin in ESUS patients, the NAVIGATE ESUS [51] and the RESPECT ESUS [52]. ESUS constitutes a heterogenous group of patients with a possible prominent role of atherosclerosis in the carotid arteries as an embolic source. Subgroup analysis [53] of the NAVIGATE ESUS provides evidence for an etiologic role of non-stenotic carotid atherosclerosis, as non-stenotic carotid plaques were much more commonly present ipsilateral to the qualifying ischemic stroke than contralateral. Similarly, a recent metanalysis [54] which combined eight studies with 323 ESUS patients, confirmed that plaques with high-risk features are five times more prevalent in the ipsilateral to the index event, compared to the contralateral ICA. In a cross-sectional observational study from Japan [55] evaluating the importance of ESUS-related atherosclerosis with CDU, ICA plaque with a thickness ≥1.5 mm was considerably more common when ipsilateral than when contralateral to the ESUS sites, especially when plaques had a thickness ≥2.6 mm. Unstable plaques such as hypoechoic, heterogeneous, ulcerated, or mobile were more frequently observed in ipsilateral strokes than in contralateral strokes, but there was no statistically significant difference between the two sets of frequencies. Similarly, in young patients with cryptogenic stroke of anterior circulation, a high plaque burden on CDU (measured by plaque length, plaque thickness and volume) was more frequent on the symptomatic side [56]. All these observations led to the proposal of a new definition (symptomatic non stenotic carotid disease—SyNC) [57] based on the assumption that non stenotic carotid disease might play a role in stroke etiology.

## 3. Vertebral Artery

Ultrasound assessment of vertebral arteries is more challenging. Vertebral artery asymmetry is very common and approximately 50% of people have a dominant left vertebral, 25% a dominant right vertebral and 25% have both vertebral arteries of similar caliber [58]. Anatomically, the vertebral artery is divided into five segments (V0–V4). V0 is usually referred as the origin of the vertebral artery. Segments V1–V3 constitute the extracranial portion of the vertebral artery and segment V4, the intracranial portion. V0 is less easily accessible due to its location behind the clavicle, with the right being more easily visualized than the left [59,60]. The first (V1) segment extends from the origin (V0) from the subclavian artery to its entry into the foramen of the transverse process of the sixth vertebrae (C6). V0/V1 is important as it is prone to hemodynamically significant lesions due to atherosclerosis or dissection. Contrary to the well-established criteria for carotid stenosis, consensus ultrasound grading criteria for vertebral stenosis are still lacking. This is partly due to the technical difficulties such as the frequent deep and posterior origin of the vertebral arteries, presence of calcified lesions, their tortuous course, or the short neck stature [61]. A systematic review by Khan et al. [62], demonstrated a relatively low sensitivity (70%) of CDU as opposed to contrast enhanced MRA and CTA in detecting a 50–99% stenosis of the vertebral arteries, but the authors included studies over 20 years old with many of them using duplex without color.

Several groups have published reference data for vertebral arteries stenosis compared to digital subtraction angiography [63,64,65]. Estimation of stenosis is based not only on the peak systolic velocity increase over a short segment, but also in comparison to the pre- and post- stenotic velocities. Optimal cut-off values of these hemodynamic parameters in evaluation of stenosis of the proximal vertebral artery in patients with symptoms of ischemia of the posterior circulation, were determined based on receiver operator characteristics (ROC) analysis (Table 3). Indirect hemodynamic signs might help for the diagnosis of VA occlusion or severe stenosis [66]. Ostial occlusion may be detected by a parvus tardus waveform in the more distal VA. Occlusion of V4 before the origin of the posterior inferior cerebellar artery (PICA) may manifest with a high resistance waveform in the proximal extracranial portion (abnormally slow systolic, but no diastolic flow velocities).

## 4. Other Non-Atherosclerotic Vessel Pathologies

### 4.1. Dissection

CDU is a useful screening tool for patients suspected harboring dissections. B-mode and color Doppler imaging may visualize abnormalities, such as stenosis or occlusion of the vessel, suggesting the presence of a dissection, even though findings assumed to be evidentiary such as hypoechoegenic increase of wall thickness, “false” lumen, hyperechogenic intimal flap (Figure 3), or pseudoaneurysm are not frequently found^55^. Indirect signs include increased or decreased pulsatility upstream or downstream of the lesion, reversed flow in the ipsilateral ophthalmic artery, cross flow through the anterior communicating artery and collateralization of flow in the A1 segment of the ipsilateral anterior cerebral artery [67]. The absence of atherosclerosis in B-mode imaging is another indirect sign that might suggest a dissection [68], especially in association with a stenosis and/or occlusion of the arterial segment usually not affected by atherosclerosis (i.e., the distal part of the ICA 2 cm or more downstream of the carotid bifurcation, V2–V4 segment of the vertebral artery).

In patients with carotid artery territory ischemia, it is shown that normal CDU can reliably exclude the presence of a dissection with a high sensitivity (up to 96%) and negative predictive value (up to 97%), particularly when coupled with transcranial doppler (TCD) [69,70]. However, in patients with only localized symptoms (isolated Horner sign, pain, etc.) sensitivity is poor and false negative ultrasound findings are high (up to 31%) [71]. This can be explained by the fact that intramural hematoma extends (in most such cases) between media/adventitia layers and rarely causes stenosis of the vessel lumen. In these cases, ultrasound of the pupil can be useful, as it shows a comparable response to light on both sides, while there is a loss of the ipsilateral cicliospinal reflex [72]. The overall accuracy of CDU in dissections of the vertebral arteries is lower and is about 86%, according to Sturzenegger et al. [73] On that grounds, diagnosis of a dissection should be further confirmed with an MRI/MRA (especially in patients with negative neurosonographic findings). CDU/TCD will help to rule out a dissection in patients with carotid territory ischemia and to assess the intracranial hemodynamics and the recanalization process. CDU/TCD also provide useful information regarding the choice of the appropriate treatment: avoid anticoagulants if the dissection extends intracranially or favor anticoagulants in case of a free-floating thrombus or presence of high intensity transient signals (HITS) despite (dual) antiplatelets [74]. Follow up sonographic studies of patients with occluded arteries due to dissection, have demonstrated a recanalization rate of up to 68% during the first six months [75,76,77].

CDU may also assist in the timely diagnosis of aortic arch dissection in the emergency room as the cause of underlying cerebral ischemia. This may have critical therapeutic implications [78] since it may avert intravenous thrombolysis that can be complicated with aortic rupture or cardiac tamponade in cases of aortic arch dissection.

### 4.2. Inflammation

Large vessel vasculitis which may present with stroke symptoms, comprise giant cell arteritis (GCA), Takayasu arteritis and rarely (isolated) idiopathic arteritis [79,80]. European League Against Rheumatism (EULAR) recommends ultrasound of the temporal arteries (with or without axillary arteries) as the first imaging test when GCA is suspected [81]. “Halo” sign is a homogenous, hypoechoic circumferential thickening of the vessel wall [82]. It is the key morphological element for the diagnosis, reflecting the inflammation of the vessel wall. “Compression” sign further confirms the presence of a suspected halo, as the thickened wall remains visible upon compression [83]. Several metanalyses [84,85,86] have demonstrated the validity of CDU in diagnosing GCA with a sensitivity ranging between 68 to 77% and a high specificity of up to 100% in the presence of bilateral halo sign. Cut off measurements of intima media thickness in order to minimize false positive results, have been suggested. The limit for a pathological halo in the temporal artery has been set at over 0.34 mm, with measurements >0.7 mm predicting a positive biopsy result [87]. Similarly, a cut off over 1.2 mm for axillary artery IMT, has been related to improved diagnostic accuracy [88]. CDU may prompt the initiation of glucocorticoid treatment and also serve as an important monitoring tool during follow up and for treatment response evaluation even in the early stages (<7 days). A recent sub-analysis [89] of the Temporal Artery Biopsy versus Ultrasound in diagnosis of GCA (TABUL study) demonstrated that the halo was significantly smaller in patients who had received more than 4 days of glucocorticoid treatment compared to those receiving up to 4 days of treatment. Takayasu arteritis is a rare disease of young adults (<40 years old), with similar to GCA ultrasound findings. It mainly affects proximal vessel cervical vessels (common carotid artery, innominate artery and subclavian artery) which can be easily identified by CDU by the typical macaroni sign [90]. Takayasu arteritis may also manifest as a subclavian steal syndrome [91], which is discussed below in more detail.

### 4.3. Fibromuscular Dysplasia

Fibromuscular dysplasia (FMD) is a nonatherosclerotic, noninflammatory vascular disease of unknown cause. FMD predominantly affects the renal and internal carotid arteries and may result in arterial stenosis, occlusion, aneurysm, or dissection [92,93]. The prognosis of FMD can be serious as it may lead to severe hypertension, renal failure or stroke.

Similarly to dissections and contrary to atherosclerosis which develops at or just beyond the carotid bifurcation, FMD is usually present in the mid and distal cervical portion of the extracranial ICA [94]. Correspondingly, an increase in Doppler velocity may occur in the distal carotid and vertebral vessels, with associated turbulence of color flow or the spectral Doppler signal. A “string-of-beads” appearance in the same arterial region is less commonly found. Severe tortuosity (S-curve) in the distal internal carotid artery especially in individuals of less than 70 years old, has been described as a novel morphological finding of FMD. A case–control study [95] of 116 FMD patients demonstrated a strong association of S-curve and FMD (odds ratio 16.86, 95% CI 3.92–72.48; *p* < 0.0001) which was also remained significant, albeit much weaker, for patients over 70 years old.

AHA FMD 2014 recommendations [96] summarize the CDU findings as follows: “…there is an increase in velocity (peak systolic velocity 250 cm/s, EDV 100 cm/s), turbulence and tortuosity in the mid to distal internal carotid artery consistent with the presence of fibromuscular dysplasia…”

In comparison to digital subtraction angiography (DSA) which is considered to be the “gold-standard” for FMD diagnosis [97,98], CDU has possibly much lower sensitivity and specificity, but is more accurate in assessing the hemodynamic significance of the lesions. Consequently, and due to the low-risk and low-cost nature of duplex ultrasound, CDU is an excellent modality for surveillance of carotid artery FMD.

## 5. Subclavian Steal Syndrome

The term subclavian steal phenomenon (SSP) describes retrograde blood flow in the vertebral artery associated with proximal (to vertebral artery origin) subclavian artery or innominate artery stenosis or occlusion [99]. In the majority of patients reversed blood flow of the vertebral artery is benign without causing any significant symptoms [100,101]. SSP does not warrant invasive evaluation or therapy, and represents an appropriate physiological response to proximal arterial disease. Subclavian steal syndrome (SSS) implies the presence of significant symptoms due to arterial insufficiency in the brain (i.e., vertebrobasilar insufficiency) or upper extremity which are supplied by the subclavian artery [102]. The most important predicting factor for symptomatic presentation of SSS is the magnitude of arterial blood pressure difference between upper extremities. SSS is linked to increased cardiovascular and all-cause mortality [103]. CDU is the first diagnostic approach with a high sensitivity (>95%) but lower specificity (85%). TCD is very useful in the setting of neurologic symptoms, as it may sufficiently assess the hemodynamic repercussion of SSS to the brain circulation [104]. Depending on the degree of the subclavian artery stenosis a systolic deceleration may be observed within the ipsilateral vertebral artery (grade 1), an alternating (biphasic) flow with the retrograde flow in the systolic phase (grade 2) or even a constant retrograde flow (grade 3) [105]. Ischemia–hyperemia cuff test provides real-time demonstration of the steal phenomenon by exacerbating the abnormal hemodynamics of a subclavian stenosis, and is specifically useful to demonstrate flow spectrum disturbances of the intracranial arteries. The cuff is inflated to more than 20 mmHg above supra-systolic levels for 1–2 min, followed by rapid deflation. Reactive hyperemia induced after release of the sphygmomanometer, results normally in higher degrees of SSS waveforms in the vertebrobasilar arteries, along with a compensatory increase of the systolic velocity of the contralateral vertebral artery [92,106,107]. The flow pattern of the brachial artery should be part of the examination in order to avoid common diagnostic pitfalls [108] related to bi-directional flow of the intracranial vertebral artery portion (V4). A normal high resistance triphasic flow pattern excludes a relevant obstruction of the proximal subclavian artery leading to a low resistance monophasic flow. SSP identified with CDU is a marker of increased cardiovascular risk that should be treated with aggressive secondary prevention similar to peripheral artery disease: antiplatelets, high dose statins, smoking cessation and tight blood pressure control. Symptomatic SSS due to hemodynamic subclavian artery lesions, warrants interventional treatment (either surgical or endovascular).

## 6. Conclusions

Stroke remains one of the leading causes of mortality and morbidity. Assessing ischemic stroke etiology and mechanism during the acute phase is crucial in order to tailor and monitor appropriate treatment and determine prognosis. Ultrasound of carotid and vertebral arteries coupled by transcranial doppler sonography are essential parts of the diagnostic workflow in every acute stroke unit. CDU provides well validated estimates of the type of atherosclerotic plaque, degree of stenosis, as well as structural and hemodynamic information about extracranial vessels (e.g., subclavian steal syndrome) and indirectly about intracranial circulation.

## Figures and Tables

**Figure 1 medicina-56-00711-f001:**
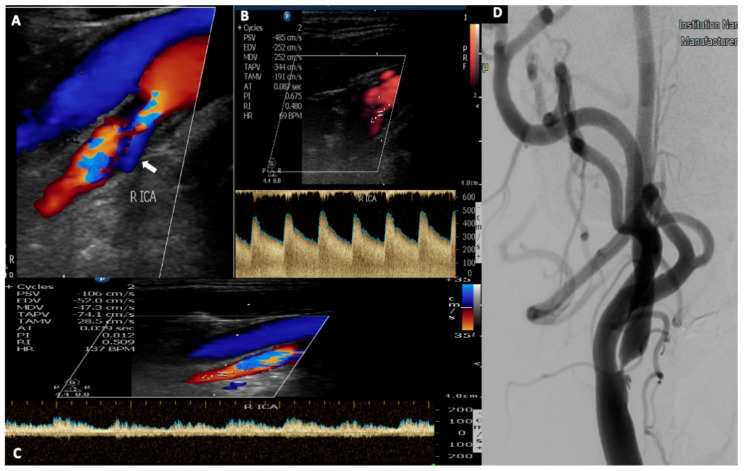
A 52-year-old patient with a history of hypertension and dyslipidemia presented with an episode of transient left arm paresis. (**A**). Urgent neurovascular ultrasound disclosed the presence of a hypoechoic plaque in his right internal carotid artery causing a severe stenosis. Note the presence of aliasing on color-mode display, the ulcer with the reversed blood flow ((**A**), white arrow) and also the elevated peak systolic and end diastolic velocity (485/252 cm/s) (**B**), which correspond to a 70–99% stenosis according to NASCET trial range. (**C**) Post stenotic Doppler spectral waveform depicts a characteristic, spiny border (wall covibration) attributed to increased flow volume. (**D**) Digital subtraction angiography performed before the carotid artery stenting, confirms the presence of a high-grade stenosis due to an ulcerated atherosclerotic plaque of the internal carotid artery.

**Figure 2 medicina-56-00711-f002:**
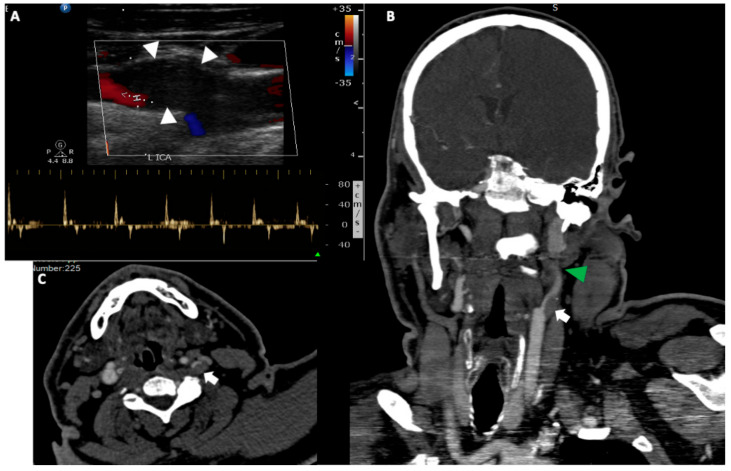
A 50-year-old man was transferred to the emergency department with right side hemiplegia, aphasia and stupor. (**A**). Urgent cervical duplex ultrasound disclosed a hypoechoic thrombus (white arrowheads) at the internal carotid artery origin, which nearly occluded the vessel lumen. Spectral analysis showed low flow velocities with small retrograde flow component and absence of end diastolic flow (42/0 cm/s), a reverberating pattern strongly suggestive of a downstream internal carotid occlusion. Coronal reformatted (**B**) and axial (**C**) CT angiography of the head and neck, confirmed the presence of a large hypodense thrombus (white arrows) at the left internal carotid artery bifurcation (the localisation aids at the differential diagnosis from a possible dissection) associated with a non-opacification of the high cervical and petrous portion of the artery due to a tandem occlusion (green arrowhead).

**Figure 3 medicina-56-00711-f003:**
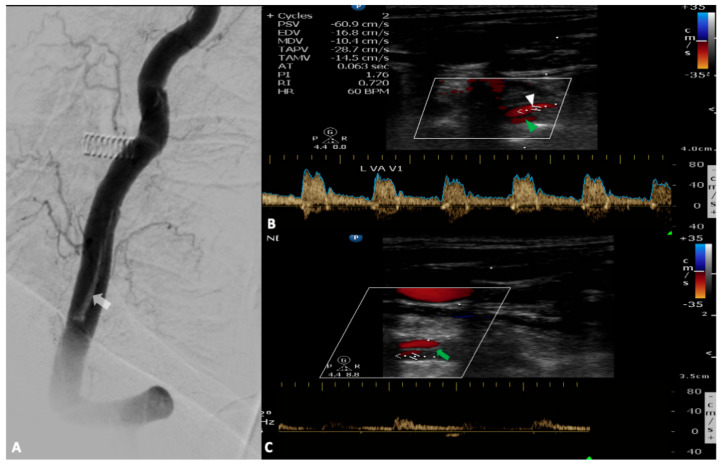
A 72-year-old woman was hospitalized for an interventional repair of a giant basilar artery aneurysm. At the end of the procedure, digital subtraction angiography revealed an intimal flap in the left vertebral artery ((**A**), white arrow). During her stay in the stroke unit she developed acute vertigo, dysarthria, left hemiparesis and opthalmoplegia. Urgent neurovascular ultrasound revealed the dissection with a normal flow in the true lumen ((**B**), white arrowhead) and a minimal flow in the pseudolumen attributed to acute thrombosis ((**C**), green arrowhead) which has the same direction with the blood flow through the true lumen (entry-exit dissection). Note the presence of the intimal flap demonstrated as a hyperechogenic strip in the middle of the vertebral artery (green arrow).

**Table 1 medicina-56-00711-t001:** Fast track neurovascular examination [6].

**A. Clinical Diagnosis of Anterior Circulation Ischemia**
**STEP 1: Transcranial Doppler**
Begin insonation in the nonaffected side, normal middle cerebral artery (MCA) waveform (M1 depth 45–65 mm, M2 30–45 mm), and velocity for comparison to the affected side. On the affected side: first assess MCA at 50 mm. If no signals detected, increase the depth to 62 mm. If an antegrade flow signal is found, reduce the depth to trace the MCA stem or identify the worst residual flow signal. Search for possible flow diversion to the anterior cerebral artery (ACA), posterior cerebral artery (PCA) or M2 MCA. Evaluate and compare waveform shapes and systolic flow acceleration. Continue on the affected side (transorbital window). Check flow direction and pulsatility in the ophthalmic artery (OA) at depths 40–50 mm followed by internal carotid artery (ICA) siphon at depths 55–65 mm. If time permits, evaluate basilar artery (BA) (depth 80–100 mm) and terminal vertebral artery (VA) (40–80 mm).
**STEP 2: Carotid/Vertebral Duplex**
Start on the affected side in transverse B-mode planes followed by color or power-mode sweep from proximal to distal carotid segments. Identify common carotid artery (CCA) and its bifurcation on B-mode and flow-carrying lumens. Document if ICA (or CCA) has a lesion on B-mode and corresponding disturbances on flow images. In patients with concomitant chest pain, evaluate CCA as close to the origin as possible. Perform angle-corrected spectral velocity measurements in the mid-to-distal CCA, internal and external carotid artery. If time permits examine cervical portion of the vertebral arteries (longitudinal B-mode, color or power mode, spectral Doppler) on the affected side. If time permits, perform transverse and longitudinal scanning of the arteries on the nonaffected side.
**B. Clinical Diagnosis of Cerebral Ischemia in the Posterior Circulation**
**STEP 1: Transcranial Doppler**
Start suboccipital insonation at 75 mm (VA junction) and identify BA flow at 80–100 mm. If abnormal signals present at 75–100 mm, find the terminal VA (40–80 mm) on the nonaffected side for comparison and evaluate the terminal VA on the affected side at similar depths. Continue with transtemporal examination to identify PCA (55–75 mm) and possible collateral flow through the posterior communicating artery (check both sides). If time permits, evaluate both MCAs and ACAs (60–75 mm) for possible compensatory velocity increase as an indirect sign of basilar artery obstruction.
**STEP 2: Vertebral/Carotid Duplex Ultrasound**
Start on the affected side by locating CCA using longitudinal B-mode plane, and turn transducer downward to visualize shadows from transverse processes of midcervical vertebrae. Apply color modes to identify flow in intratransverse VA segments. Follow VA course to its origin and obtain Doppler spectra. Perform similar examination on another side. If time permits, perform bilateral duplex examination of the CCA, ICA, and external carotid artery as described above.

**Table 2 medicina-56-00711-t002:** Ultrasound diagnostic criteria for reporting carotid ultrasound investigations [44].

Percentage Stenosis (NASCET)	ICA Peak Systolic Velocity (cm/s)	Peak Systolic Velocity Ratio ICA_PSV_/CCA_PSV_	St Mary’s Ratio ICA_PSV_/CCA_EDV_
<50%	<125	<2	<8
50–59%	>125	2–4	8–10
60–69%			11–13
70–79%	>230	>4	14–21
80–89%			22–29
>90 but less than near occlusion	>400	>5	>30
Near Occlusion	High, low-string flow	Variable	Variable
Occlusion	No flow	Not Applicable	Not Applicable

**Table 3 medicina-56-00711-t003:** Reported cut-off values of hemodynamic parameters in evaluation of stenosis of the proximal vertebral artery [60].

Percentage Stenosis	PSV_origin_ (cm/s)	Ratio PSV_origin_/PSV_IV_	EDV_origin_ (cm/s)
<50%	≥85	≥1.3	≥27
50–69%	≥140	≥2.1	≥35
70–99%	≥210	≥4.0	≥50

PSV_origin_: Peak Systolic Velocity at the origin of the vertebral artery; PSV_IV_: Peak Systolic Velocity in intravertebral segments; EDV_origin_: End Diastolic Velocity at the origin of the vertebral artery.

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
