# Peer review of "Ultrasound Assessment of Extracranial Carotids and Vertebral Arteries in Acute Cerebral Ischemia"

_medicina, 2020, doi:10.3390/medicina56120711_

Round 1
Reviewer 1 Report
This is a well-organized, adequately-balanced and well-written overview of the important role of ultrasound of extracranial carotids and vertebral arteries in acute cerebral ischemia. In times of COVID-19 pandemic, the use of a non invasive bedside tool is very helpful to the stroke clinician and therefore the authors should be praised for this work.
I would just suggest adding the following:
Page 8 line 272: …media/adventitia layers and rarely causes stenosis of the vessel lumen. In these cases, ultrasound of the pupil can be useful, as it shows a comparable response to light on both sides, while there is a loss of the ipsilateral cicliospinal reflex (Farina F, Brunner C, Schreiber SJ, Palmieri A, Struhal W, Baracchini C, Vosko MR. Ultrasound examination of the pupil suggestive for carotid dissection. Neurology. 2017 Aug 29;89(9):973-974). The overall accuracy of CDU…….
Author Response
The Reviewers raised some excellent points, which we were happy to incorporate into the revised manuscript. A point-by-point reply to each comment is provided below. We take this opportunity to thank the reviewers and editorial staff for their time and consideration.
Reviewer #1
General comment
“This is a well-organized, adequately-balanced and well-written overview of the important role of ultrasound of extracranial carotids and vertebral arteries in acute cerebral ischemia. In times of COVID-19 pandemic, the use of a non invasive bedside tool is very helpful to the stroke clinician and therefore the authors should be praised for this work..”
Response: We would like to thank Reviewer #1 for summarizing so concisely our work and the positive feedback she or he had provided regarding our manuscript. No changes were made regarding the aforementioned general comments.
Comment 1
“Page 8 line 272: …media/adventitia layers and rarely causes stenosis of the vessel lumen. In these cases, ultrasound of the pupil can be useful, as it shows a comparable response to light on both sides, while there is a loss of the ipsilateral cicliospinal reflex (Farina F, Brunner C, Schreiber SJ, Palmieri A, Struhal W, Baracchini C, Vosko MR. Ultrasound examination of the pupil suggestive for carotid dissection. Neurology. 2017 Aug 29;89(9):973-974).”
Response: We thank the reviewer for this comment. This novel information has been added in the revised manuscript (page 8 line 380-381).

Reviewer 2 Report
The authors submit a narrative, not systematic, review about the applications of Cervical Duplex Ultrasound in acute stroke management. The text and the logic are clear. The abstract is informative and reflects the body of the paper. The review does not provide an original contribution to the field but it is well done. It may represent a useful tool for those who want to approach the subject.
I find that the description of the different topics has an insufficient correlation with the acute phase. I suggest adding in the different parts the consequences in clinical practice in terms of therapeutic choices, as the authors themselves announce in the abstract.
I also suggest illustrating how CDU should be integrated in the acute stroke workflow. For example, it may be helpful to illustrate methods of "Fast-Track Neurovascular Ultrasound Examination".
In conclusion, the review is informative, but the paper requires some revision.
Author Response
The Reviewers raised some excellent points, which we were happy to incorporate into the revised manuscript. A point-by-point reply to each comment is provided below. We take this opportunity to thank the reviewers and editorial staff for their time and consideration
Reviewer #2
General Comment
“The authors submit a narrative, not systematic, review about the applications of Cervical Duplex Ultrasound in acute stroke management. The text and the logic are clear. The abstract is informative and reflects the body of the paper. The review does not provide an original contribution to the field but it is well done. It may represent a useful tool for those who want to approach the subject.”
Response: We would like to thank Reviewer #2 for her/his insightful comments and her/his careful assessment of our manuscript. No changes were made regarding the aforementioned general comment.
Comment 1
“I find that the description of the different topics has an insufficient correlation with the acute phase. I suggest adding in the different parts the consequences in clinical practice in terms of therapeutic choices, as the authors themselves announce in the abstract.”
Response: We appreciate reviewer 2 for this suggestion. We would like to clarificate that our intention was not only to provide the importance of carotid/vertebral ultrasound during the hyperacute phase when revascularization therapies are considered but also during the whole acute period of a stroke unit hospitalization which integrated the diagnosis and the initiation of secondary prevention therapies. We have now added some important points, that highlight the potential therapeutic implications of CDU:
Page 5, line 149-152:
“In patients with a TIA or mild stroke and less than 50% stenosis, dual antiplatelet therapy (DAPT) with a loading aspirin and clopidogrel dose (or alternatively ticagrelor) in the first 24 hours after presentation, is typically recommended for the first 21-30 days.”
Page 9, line 289-292:
“CDU/TCD also provide useful information regarding the choice of the appropriate treatment: avoid anticoagulants if the dissection extends intracranially or favor anticoagulants in case of a free-floating thrombus or presence of high intensity transient signals (HITS) despite (dual) antiplatelets”
Page 9, line 312-314:
“CDU may prompt the initiation of glucocorticoid treatment and also serve as an important monitoring tool during follow up and for treatment response evaluation even in the early stages (<7 days).”
Page 10 line 370-374
“SSP identified with CDU is a marker of increased cardiovascular risk that should be treated with aggressive secondary prevention similar to peripheral artery disease: antiplatelets, high dose statins, smoking cessation and tight blood pressure control. Symptomatic SSS due to hemodynamic subclavian artery lesions, warrants interventional treatment (either surgical or endovascular).”
Comment 2
“I also suggest illustrating how CDU should be integrated in the acute stroke workflow. For example, it may be helpful to illustrate methods of "Fast-Track Neurovascular Ultrasound Examination”
Response: Thank you for the comment. We have now added a new table illustrating a fast track neurovascular ultrasound examination protocol with the appropriate bibliographic reference.
Page 2 line 65-67
“A fast-track neurovascular ultrasound examination with high accuracy in detecting arterial lesions, can be integrated in the hyperacute stroke workflow and facilitate revascularization therapies (table 1).”
Let us again emphasize that we appreciate the Reviewers and the Editorial Board for the careful attention that our manuscript received and that in our resubmission we were happy to address their constructive comments and criticism.
